# Knowledge Distillation for High Dimensional Search Index

**Zepu Lu**[1,2], **Jin Chen**[3], **Defu Lian**[1,2]*, **Zaixi Zhang**[1,2], **Yong Ge**[4], **Enhong Chen**[1,2]

[1]School of Computer Science and Technology, University of Science and Technology of China
[2]State Key Laboratory of Cognitive Intelligence, Hefei, Anhui, China
[3]University of Electronic Science and Technology of China
[4]University of Arizona
zplu@mail.ustc.edu.cn, chenjin@std.uestc.edu.cn, liandefu@ustc.edu.cn, zaixi@mail.ustc.edu.cn,
yongge@arizona.edu, cheneh@ustc.edu.cn

## Abstract

Lightweight compressed indexes are prevalent in Approximate Nearest Neighbor Search (ANNS) and Maximum Inner Product Search (MIPS) owing to their superiority of retrieval efficiency in large-scale datasets. However, results given by compressed indexes are less accurate due to the curse of dimension and limitation of optimization objectives (e.g., lacking interactions between queries and documents). Thus, we are encouraged to design a new learning algorithm for the compressed search index in high dimensions to improve retrieval performance. In this paper, we propose a novel **K**nowledge **D**istillation for high dimensional search **index** framework (**KDindex**), with the aim of efficiently learning lightweight indexes by distilling knowledge from high-precision ANNS and MIPS models such as graph-based indexes. Specifically, the student is guided to keep the same ranking order of the top-k relevant results yielded by the teacher model, which acts as the additional supervision signals between queries and documents to learn the similarities between documents. Furthermore, to avoid the trivial solutions that all candidates are partitioned to the same post list, the reconstruction loss that minimizes the compressed error, and the posting list balance strategy that equally allocates the candidates, are integrated into the learning objective. Experiment results demonstrate that **KDindex** outperforms existing learnable quantization-based indexes and is 40× lighter than the state-of-the-art non-exhaustive methods while achieving comparable recall quality.

## 1 Introduction

Vector nearest neighbor search, which retrieves the most relevant vectors with the maximum similarity given the query vector, is a fundamental task in information retrieval, such as image retrieval [13, 55], web search [51, 52, 53], and item recommendation [32, 5, 33, 35, 48]. With a significant number of candidate vectors and vector dimensions, the exact nearest neighbor search becomes intractable due to the substantial computation costs and high query latency [42], resulting in a growing interest in Approximate Nearest Neighbors Search (ANNS).

Among the various indexes, the lightweight compressed indexes, especially those based on quantization, are notable for their substantial advantages in terms of low storage costs and efficient parallel processing. As a result, these indexes draw the attention from both academics and industries and are present in well-known open repositories such as FAISS [26]. Existing learning methods for

---

*Corresponding Author

37th Conference on Neural Information Processing Systems (NeurIPS 2023).

quantizers differ in whether there are explicit labels and whether they are related to queries. Traditional clustering-based methods for learning codebooks, e.g., PQ [24], OPQ [18], and AQ [3], are categorized as un-supervised algorithms only connected with document embeddings without the query information. Another type of technique [20] involves training the codebooks by optimizing the reconstruction-based loss, which relies on correlations with queries and documents but lacks explicit natural supervision signals. The last category of algorithms employs the query-dependent ground-truth labels to improve the retrieval performance, such as the existing interactions between query and candidates in LightRec [34], the ground-truth nearest neighbors in BLISS [21]. These additionally introduced supervision signals help capture the relationships between items and queries, thus contributing to better quantizers. However, the interaction label is only available in a small set of datasets and the expense of obtaining the ground-truth labels is particularly high, making the scenarios without any label information a more common case.

In this paper, we creatively propose a **K**nowledge **D**istillation for high dimensional search **index** framework (**KDindex** for short) to distill knowledge from a more advanced teacher index model into lightweight indexes, under the circumstances without available label information. The advanced indexes, such as graph-based indexes, can yield more accurate results owing to their powerful expressiveness of the vector space and hence becomes able to teach the less-accurate lightweight indexes. Specifically, the top-k nearest results obtained from the teacher indexes act as the supervision signals to optimize the compressed functions. Therefore, index distillation is formulated as the top-k learning for the student index model from the teacher index. Specific ranking-oriented losses are exploited to learn the knowledge from the teacher model, which guides the results from the student model to have the same ranking orders as the teacher models. To avoid the trivial solutions, where the candidate vectors collapse to the identical centroid, we have two learning constraints for the learning objective. The reconstruction loss minimizes the distance between the original input and the compressed input, which pushes the centroids by the different query and candidate vectors. The second strategy is the posting list balance, which encourages an equal distribution of documents among each centroid. Thus, the candidates would be equally assigned to different centroids to avoid trivial solutions. Lastly, unlike recent iterative approaches that compulsively modify indexes, our approach uses a differentiable training process that updates the centroids and indexes simultaneously per mini-batch. This aims to minimize the error resulting from asynchronous updates. In this way, the student model would yield better performance thanks to the powerful expressiveness of the neighbor relationships of the teacher model, while keeping the low inference time and storage cost.

The contribution of this work can be summarized as follows:

- To the best of our knowledge, this is the first attempt to extract knowledge from the high-precision search index with advanced structures into lightweight indexes in order to improve the retrieval performance of the lightweight index in high dimensions.

- We propose the index distillation paradigm to learn the top-k nearest neighbors retrieved from the teacher models, where the top-k retrieved results act as the supervision signals to guide the learning of centroids, which is label-free and takes the query information into consideration.

- We utilize ranking-oriented objectives as distillation loss with two learning constraints to avoid trivial solutions. Furthermore, we design the differentiable training process to avoid the asynchronous update of centroids and index assignments.

- Experimental results on four benchmark datasets demonstrate that KDindex achieves a 40x index compression ratio, and 2x CPU speedup while maintaining comparable retrieval performance as the state-of-the-art compressed ANNS models.

## 2 Preliminaries

### 2.1 Problem Definition

Assume that there are $M$ candidates $\{i_j\}_{j=1}^M$, such as documents, images, and items, in the retrieval system with no additional interaction information and the $i$-th candidate is represented by a $D$-dimensional vector $\boldsymbol{d}_i \in \mathbb{R}^D$. $D$ usually appears to be huge numbers (larger than 100) in nowadays retrieval systems. Given a query vector $\boldsymbol{q} \in \mathbb{R}^D$, the system attempts to return the top-k relevant candidates depending on the similarity scores $S(\cdot, \cdot)$ between the query and candidate vectors, such as the inner product, L2 distance, and cosine distance.

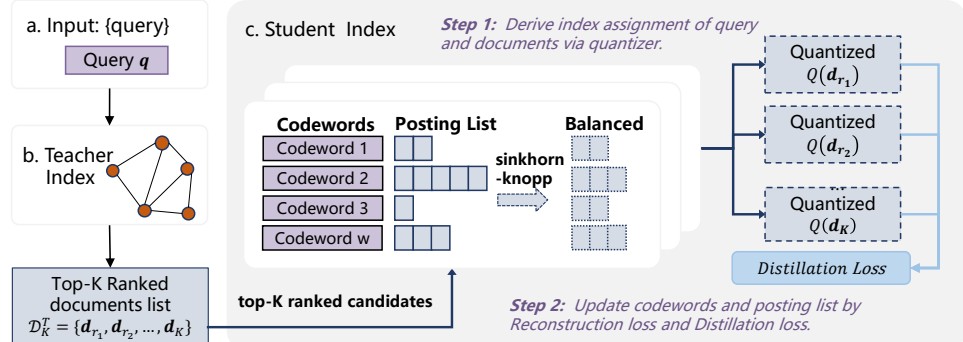

Figure 1: Illustration of KDindex framework. Given the teacher index built on full-corpus, top-K ranked documents are returned for each query vector. Then, the query and corresponding documents are compressed by lightweight indexes (such as PQ, OPQ, and AQ) and are ideally encoded with similar indexes, which are guided by the ranking-oriented distillation loss. Reconstruction loss and Sinkhorn-Knopp are used to avoid trivial solutions.

## 2.2 Lightweight Compressed Indexes

With the increasing number of candidates, Approximate Nearest Neighbors Search (ANNS) and Maximum Inner Product Search (MIPS) strategies are widely used to accelerate the search speed at the expense of accuracy, among which lightweight compressed indexes (hashing and quantization) are widely used because of their strong performance in terms of both effectiveness and efficiency.

The quantization-based indexes, a representative compressed ANNS algorithm, quantize each candidate vector into the reconstruction vector from the $B$ codebooks, each of which contains $W$ vectors (codewords). We denote the $b$-th codebook as $C^b$ and the $w$-th codeword in the $b$-th codebook as $c_w^b$. The Additive Quantization (AQ) encodes the vector as a sum of $B$ codewords, i.e., $\tilde{d} = Q(d) = \sum_{b=1}^{B} c_{w_b}^b$, while the Product Quantization (PQ) and Optimized product quantization (OPQ) encode the vector as the concatenation of $B$ codewords of each subspace, i.e., $\tilde{d} = Q(d) = c_{w_1}^1 \oplus c_{w_2}^2 \oplus ... \oplus c_{w_B}^B$. The codeword with the minimal distance from the original vector is chosen, i.e., $c_w^b = \underset{i \in \{1,2,...,W\}}{\arg\min} \|x - c_{w_i}^b\|^2$. $Q(\cdot)$ denotes the quantizer function. The codewords are usually learned by the unsupervised clustering algorithms or the reconstruction loss, and the posting lists which record the belonging candidates are then constructed. During the inference stage, the similarity scores between the query vector and the codewords are calculated and the approximate nearest candidates are oriented. Consequently, given the well-trained indexes, the overall inference time complexity is only correlated with the number of codewords and codebooks.

Learning to hash is also an effective method to compress high-dimensional data into low-dimensional binarized codes, similar to those clustering-based conventional quantization methods. Learning to hash approaches have the property that objects that are close to each other have a higher probability of colliding than objects that are far apart across various distance metrics. The drawbacks of these approaches are the requirement for a large number of hash tables in order to achieve good search quality and these methods are unmindful of the distribution of vectors, often leading to lop-sided partitions and long query times. Thus, we focus more on the quantization-based compressed methods.

## 3 Index Distillation

Despite the efficiency advantages of lightweight quantization-based indexes, they are still limited by the representation space of codewords. Existing works introduce additional ground-truth labels to supervise the learning. For example, LightRec [34] exploits the interaction information between the certain query and document, and BLISS [21] takes the ground-truth neighbors over candidates to enhance the discrimination probability. These extra signals benefit better capturing the query-dependent similarity relationships between candidates. However, in more circumstances, there are no ground-truth labels considering the high expense to obtain them. Thus, we focus on scenarios without label information and aim to enhance the performance of lightweight indexes.

Several existing works introduce the knowledge of query to adjust the choice of the centroids [20] or to randomly sample neighbors to learn the centroids. These works provide less relevant candidates, thus resulting in less accurate inference results. Intuitively, if we use more progressive index structures, such as graph-based methods, to encode the similarity information over the whole vector space, more accurate results would be obtained. This motivates the progressive indexes as the teacher to distill more informative high-dimensional knowledge to the lightweight quantization-based indexes, with the goal of improving the accuracy of lightweight indexes under high dimensions.

## 3.1 Overview of Index Distillation

The whole framework, i.e., Knowledge Distillation for high dimensional search index (KDindex), consists of a high-precision teacher search model, which is well-learned, and a lightweight student search model to be learned, where the student model attempts to learn knowledge from the top-k relevant candidates retrieved from the teacher model. Specifically, given the well-trained candidate vectors, when a query vector $q$ requests, the teacher search model returns the set of the top-k approximate nearest neighbors $\mathcal{D}_K^T = \{i | rank(i|q) \leq K\}$. These candidates then behave as weak signals to supervise the learning of codewords in the student models, where the student indexes are guided to learn the same ranking orders and finally to return the similar indexes for the given query and candidate. Figure 1 visualizes the whole training process of KDindex, with the three key components: Distillation Loss, two Learning Constraints, and a Differentiable Training paradigm which are detailed in the following sections. The whole training process is illustrated in Algorithm 2.

## 3.2 Initialization

The initialization of the student search model involves the following steps: (1) Initialize the centroids (codewords) and assign indexes (2) Update the centroids based on the distillation loss and reconstruction loss given the training queries (3) Adjust the posting lists depending on the latest centroids. A straightforward process is to update the posting lists per $T$ epochs after centroids updating is finished by enumerating all training queries, similar to the paradigm of BLISS [21].

## 3.3 Distillation Loss

Knowledge distillation was first proposed for classification tasks [22], where the probabilities of each class attained from the large-scale teacher network are considered as soft labels to supervise the learning of the small-size student network. The cross-entropy loss is commonly used as the distillation loss to minimize the difference between the teacher and student networks. Here, the teacher search model provides the top-k relevant candidates rather than the continuous value of probabilities. We exploit the following **Distributed-based loss** as the distillation loss to guide the student indexes to return the same nearest results. This objective function follows as:

$$\mathcal{L}(q, \mathcal{D}_K^T; C) = -\sum_{i \in \mathcal{D}_K^T} \sum_{b=1}^{B} \sum_{k=1}^{W} \tilde{p}_{bk}^q \log(\tilde{p}_{bk}^{d_i} \cdot w_i) \tag{1}$$

where $B$ denotes the number of codebooks and $W$ is the number of codewords in each codebook. $w_i = \frac{1}{rank(i)}$ corresponds to the top-k list given from the teacher model. $p_{bk}^q$ denotes the similarity score between the query $q$ and the codeword $c_k^b$, i.e., $p_{bk}^q = S(q, c_k^b)$, and $p_{bk}^{d_i}$ denotes the similarity score between the candidate $d_i$ and the codeword $c_k^b$, i.e., $p_{bk}^{d_i} = S(d_i, c_k^b)$. The normalized value $\tilde{p}_{bk}^q$ and $\tilde{p}_{bk}^{d_i}$ are calculated over the $W$ codewords for each codebook through the softmax function.

This loss attempts to minimize the distance between the queries and top-k neighbors by calculating the similarity scores with all the centroids. Thus, we could obtain more information from centroids and focus on the top-K nearest neighbors.

## 3.4 Learning Constraints

Although the distillation loss supervises the learning of quantizers, the student model would collapse with a trivial solution where all candidates are assigned to the identical centroid. To avoid trivial solutions, we propose two strategies and we will introduce them one by one.

**Reconstruction Loss.** A straightforward solution is to design the reconstruction loss to draw the distance between the input vector and the reconstructed vector encoded through the quantized function. Specifically, MSE loss is adopted here to specialize the reconstruction loss for the both query and candidate vectors:

$$\mathcal{L}_{\text{reconstruction}}(\boldsymbol{x}; \boldsymbol{C}) = \frac{1}{D}\|\boldsymbol{x} - Q(\boldsymbol{x})\|^2 \tag{2}$$

where $\boldsymbol{x} \in \mathbb{R}^D$ represents the original input vector such as the query vector $\boldsymbol{q}$ and the candidate vector $\boldsymbol{d}$. $Q$ denotes the quantized function to reconstruct the input vector and the corresponding codewords would be updated through the loss.

**Balanced Posting List.** The posting list refers to the candidate list according to the certain centroids. The trivial solutions would result in an imbalanced posting list, where some posting lists would be substantially longer than others and thus resulting linear complexity in the worst case. Thus, we impose a balanced clustering constraint to guide the candidate vectors to be equally assigned to all quantization centroid embeddings. Here, we take one codebook as an example to formulate the learning objective with the constraint:

$$\min_{\boldsymbol{C}} \sum_{i=1}^{M} \sum_{k=1}^{W} q(k|\boldsymbol{d}_i)\|\boldsymbol{c}_w - \boldsymbol{d}_i\|^2$$

$$\text{subject to } q(k|\boldsymbol{d}_i) \in \{0,1\}, \sum_{k=1}^{W} q(k|\boldsymbol{d}_i) = 1, \sum_{i=1}^{M} q(k|\boldsymbol{d}_i) = \frac{M}{W} \tag{3}$$

where $q(k|\boldsymbol{d}_i)$ denotes the binary value indicating whether the candidate vector is quantized to the $k$-th centroid and the second condition ensures that only one centroid is selected. The last condition guides each centroid to be allocated an equal number of candidates to achieve a balanced posting list. Thus, the indexes for different candidates would be varied to avoid the trivial solution.

To solve Eq (3), we relax the binary constraint $q(k|\boldsymbol{d}_i)$ to the continuous probability, i.e., $q(k|\boldsymbol{d}_i) \in (0,1)$, to get the approximated solution. The problem then can be reduced to the optimal transport problem by taking the distance as the cost of quantization and we use the Sinkhorn-Knopp algorithm [10] to solve it. Algorithm 1 details the balance strategy to uniformly allocate the candidates.

---

**Algorithm 1:** Posting List Balance

    **Input:** Document Vectors $\{\boldsymbol{d}_i\}_{i=1}^{M}$, Codebook $\boldsymbol{C} = \{\boldsymbol{c}_k\}_{k=1}^{W}$
    **Output:** Posting Lists $\{\mathcal{P}_k\}_{k=1}^{W}$
**1** Initialize the posting lists with empty sets ;
**2** Generate the probability matrix $\boldsymbol{P} \in \mathbb{R}^{M \times W}$ with the normalized similarity scores over the
    codewords $p_{ik} = \frac{\exp S(\boldsymbol{d}_i, \boldsymbol{c}_k)}{\sum_{k'=1}^{W} \exp S(\boldsymbol{d}_i, \boldsymbol{c}_{k'})}$ ;
**3** Get the transferred probability matrix $\boldsymbol{P}^s \in \mathbb{R}^{M \times W}$ by inputting $\boldsymbol{P}$ into *Sinkhorn-Knopp*;
**4** **for** $i \in \{1, ..., M\}$ **do**
**5**     Determine the index according to the transferred probability, i.e.,
        $t = \arg\max(p_{i1}^s, p_{i2}^s, ..., p_{iW}^s)$ ;
**6**     Add $i$ into the corresponding posting list $\mathcal{P}_t = \mathcal{P}_t \cup \{i\}$;
**7** **end**

---

### 3.5 Differentiable Training

The re-assignment of the indexes, including "$\arg\min$" operation, avoids the consecutive computation and thus interrupts the backward of the gradient, resulting in the **iterative training** with the two separate steps. However, the update of the index is later than the update of embedding, resulting in a large deviation in the quantized vector and inaccuracy of the retrieval performance. To keep the index assignment up-to-date during each training batch, we attempt to design a **differentiable training** manner for continuous computation. The Gumbel-Softmax Trick [23] is a common and popular solution, motivated by which we adopt a similar sample operation to copy the gradient into the one-hot embeddings. Surprisingly, the probabilities of each centroid are calculated in the

aforementioned balanced posting list and can be regarded as the weight of the centroids for updating, as shown in Line 9-11 of Algorithm 2.

The whole training process is illustrated in Algorithm 2, which can be computed in parallel for each mini-batch. Accordingly, the learning of centroids and the index re-assignment are updated within the same mini-batch, rather than the asynchronous epochs.

---

**Algorithm 2:** KDindex with Differentiable Training

**Input:** Training Queries $\mathcal{Q} = \{\boldsymbol{q}_m\}_{m=1}^{N}$, Document Vectors Document Vectors $\{\boldsymbol{d}_i\}_{i=1}^{M}$, Teacher Search Model $T$
**Output:** Student Search Model including Codebooks and Index assignment

1   Initialize the codebooks $\boldsymbol{C}$;
2   **while** *not reaching convergence conditions* **do**
3      Get the quantized vector for the query $Q(\boldsymbol{q})$;
4      Sample a mini-batch queries $\mathcal{Q}_B$ from $\mathcal{Q}$;
5      Assign indexes according to Algorithm 1;
6      **for** $\boldsymbol{q} \in \mathcal{Q}_B$ **do**
7          Retrieve the top-$K$ nearest neighbors $\mathcal{D}_K^T$ from the teacher search model $T$;
8          **for** $i \in \mathcal{D}_K^T$ **do**
9              Get the one hot embedding of the index, i.e., $\boldsymbol{e}_i = \text{One\_Hot}(t), t = \arg\max(p_{i1}^s, p_{i2}^s, ..., p_{iW}^s)$;
10             Copy the corresponding probability $\boldsymbol{p}_i = [p_{i1}, p_{i2}, ..., p_{iW}]$ ;
11             Update the one hot encoding with gradient, i.e., $\tilde{e}_i = (\boldsymbol{e}_i - \boldsymbol{p}_i).detach() + \boldsymbol{p}_i$ ;
12             Get the quantized vector $Q(\boldsymbol{d}_i) \approx \tilde{e}_i \cdot \boldsymbol{C}$ ;
13          **end**
14          Update the codebook $\boldsymbol{C}$ depending on the distillation loss Eq (6) and the reconstruction loss Eq (2);
15      **end**
16   **end**

---

### 3.6 Complexity Analysis

The training ways of KDindex and basic Qutization methods (AQ, PQ and OPQ) are different, therefore the complexity is different in the training phase. The indexing and inference way of KDindex and Qutization methods are consistent and there is no additional time complexity in the index and inference stages. More details are described in Table 1.

Table 1: The time complexity of KDindex, HNSW, and ScaNN. Denoted by $D$ item embedding dimension, $B$ the number of subspaces, $W$ the number of centroids in each codebook, $M$ the batch size (the number of queries in each batch), and $K$ the number of neighbors. $N$ is the number of items. As for ScaNN, $K_v$ denotes the number of centroids in VQ ( vector quantization ) and $K_p$ in PQ ( Product Quantization ).

| Methods | Initialization | Training | Indexing |
|---|---|---|---|
| KDindex (PQ) | $O(MWD)$ | $O(MWD + MBW + MBWK)$ | $O(NWD)$ |
| KDindex (OPQ) | $O(MWD^2)$ | $O(MWD^2 + MBW + MBWK)$ | $O(NBW((D/B) + D^2))$ |
| KDindex (AQ) | $O(MBWD)$ | $O(MBWD + MBW + MBWK)$ | $O(NBWD)$ |
| HNSW | N/A | N/A | $O(ND \log N)$ |
| ScaNN | N/A | $O(MWD + MBW + MBWK)$ | $O(N(BK_p(D/B) + K_vD))$ |

## 4   Experiment

### 4.1   Experiment Setup

**Datasets and Metrics.** Four large-scale retrieval benchmarks, including *SIFT1M*, *GIST1M* from ANN datasets [2], *MS MARCO Doc* and *MS MARCO Passage* from the TREC 2019 Deep Learning

Table 2: Comparison with Quantization-based methods. The improvement over the strongest baseline is statistically significant on a paired t-test ($p < 0.05$).

| Model | SIFT1M Recall@10 | GIST1M Recall@10 | MS MARCO Doc Recall@10 | MS MARCO Passage Recall@10 |
|---|---|---|---|---|
| PQ [24] | 31.27±0.12 | 5.04±0.14 | 2.85±0.32 | 1.80±0.08 |
| OPQ [18] | 33.85±0.14 | 15.99±0.17 | 14.63±0.25 | 9.13±0.53 |
| AQ [3] | 35.48±0.34 | 19.32±0.19 | 16.73±0.57 | 10.18±0.69 |
| DiffPQ [6] | 29.01±0.24 | 4.43±0.29 | 4.17±0.26 | 2.53±0.06 |
| DeepPQ [17] | 25.02±0.50 | 4.39±0.18 | 6.47±0.33 | 8.43±0.72 |
| GCD [25] | 33.58±0.63 | 15.76±0.13 | 15.59±0.27 | 10.28±0.46 |
| RepCONC [53] | 33.59±0.32 | 15.32±0.46 | 16.07±0.31 | 10.37±0.37 |
| PQVAE [49] | 30.39±0.65 | 6.09±0.33 | 12.43±0.25 | 8.26±0.21 |
| KDindex (PQ) | 32.53±0.34 | 9.43±0.15 | 8.74±0.32 | 4.32±0.53 |
| KDindex (OPQ) | 34.77±0.47 | 17.32±0.17 | 16.70±0.28 | 10.66±0.69 |
| KDindex (AQ) | 37.30±0.17 | 21.33±0.23 | 18.93±0.76 | 11.19±0.26 |
| | NDCG@10 | NDCG@10 | MRR@10 | MRR@10 |
| PQ | 73.21±0.17 | 19.84±0.75 | 3.82±0.19 | 2.84±0.35 |
| OPQ | 75.76±0.09 | 49.10±0.74 | 34.75±0.14 | 29.06±0.49 |
| AQ | 77.82±0.42 | 60.33±0.63 | 38.52±1.02 | 33.52±0.58 |
| DiffPQ | 70.53±0.16 | 17.39±0.64 | 10.43±0.39 | 3.69±0.12 |
| DeepPQ | 65.80±0.75 | 16.60±0.71 | 14.90±0.18 | 28.33±0.73 |
| GCD | 75.49±0.23 | 48.82±0.58 | 38.89±0.29 | 34.01±0.62 |
| RepCONC | 75.47±0.52 | 47.69±0.79 | 39.03±0.73 | 34.27±0.59 |
| PQVAE | 69.84±0.27 | 23.48±0.42 | 34.47±0.92 | 27.91±0.74 |
| KDindex (PQ) | 73.90±0.59 | 30.62±0.58 | 17.64±0.13 | 6.42±0.59 |
| KDindex (OPQ) | 76.32±0.63 | 52.36±0.49 | 39.75±0.25 | 34.62±0.47 |
| KDindex (AQ) | 80.01±0.37 | 63.17±0.62 | 41.69±0.34 | 35.23±0.38 |

Track [9], are used to validate the effectiveness of the proposed KDindex. SIFT1M and GIST1M both have exactly 1 Million database points, and the dimension is 128 and 960, respectively. Document Retrieval consists of $3.2M$ documents, $0.36M$ training queries, and $5K$ development queries. Passage Retrieval has a corpus of $8.8M$ passages, $0.8M$ training queries, and $0.1M$ development queries. All the datasets offer the set of training queries and testing queries. The detailed specifications of the datasets are shown in Table 6. The search model returns the top-$K$ retrieval results given the test query and thus we use ranking-based metrics to evaluate the performance, including Recall, NDCG and MRR with a cutoff of 10.

**Implementation Details.** SIFT1M and GIST1M datasets contain the candidate vectors with the dimension of 128 and 960 respectively. The vectors for MS MARCO Doc and MS MARCO Passage are generated based on huggingface transformers [47] and the dimension is set to 768 following previous works [14]. The similarity function $S$ for SIFT1M and GIST1M is the Euclidean distance and for MS MARCO Doc and Passage is the inner product. We exploit a well-trained HNSW [37] model as the teacher model and $K = 10$ nearest neighbors are retrieved accordingly. We adopt PQ, OPQ, and AQ as the student model for KDindex. For a fair comparison, we run current supervised methods with optimized objectives proposed in KDindex instead of interaction information-based loss function. Each vector is quantized by $B = 8$ codebooks, each of which contains $W = 256$ codewords by default. The centroids are trained with a learning rate of 0.01 and optimized by the Adam [28] optimizer. The batch size is set to 64.

## 4.2 Overall performance

We first compare the retrieval performance within the same retrieval time to verify the effectiveness of KDindex. Experiments are run 5 times to conduct the t-test and the performance of the retrieval is illustrated in Table 2, from which we have the following findings:

*KDindex respectively improves the retrieval performance of three native student models, demonstrating the general effectiveness of KDindex.* KDindex has a relative 10.49% and 9.65% improvement in

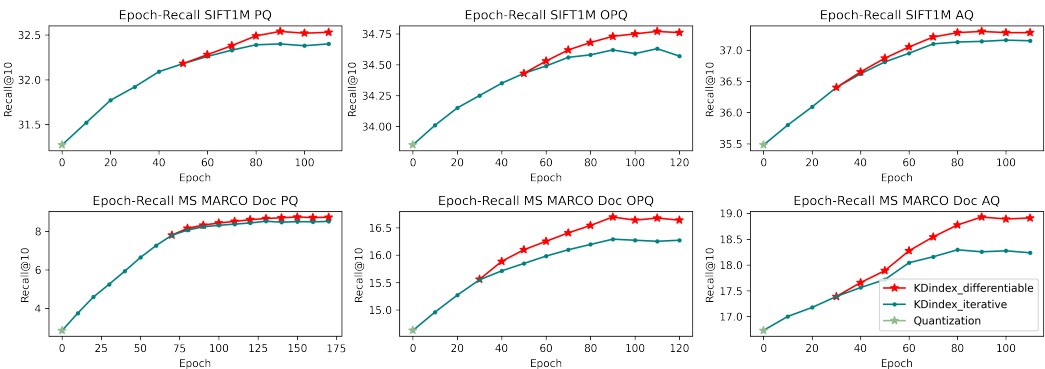

Figure 2: Curves for recall during training warmed up by initialization.

Recall@10 over all datasets compared with the native OPQ and AQ, respectively. This demonstrates that KDindex receives the supervision signals of the top-k retrieved results and thus more neighbour relationships are distilled for the student, resulting in better performance of item recall.

*KDindex shows better performance than learnable quantization-based indexes with no ground-truth labels.* By choosing appropriate quantization-based indexes, KDindex would outperform all baseline methods, especially the learnable indexes DeepPQ, demonstrating the advantage of the proposed KDindex. The introduction of distillation loss has an edge over capturing the relationships between candidates, where the top-k retrieved candidates provide stronger signals for learning.

*The improvement of KDindex is more significant when the distribution between query space and database space is different.* KDindex obtains more improvements on MS MARCO Doc and MS MARCO Passage datasets which search documents by the inner product. The similarity could not be obtained by original quantization methods. Experiments have proved that KDindex could indeed learn the neighbor relationship between query vectors and candidates by the distillation loss function.

### 4.3 Performance of Differentiable Training

It is extremely difficult for the model to learn the codebooks as well as the index at the same time during the initialization phase in a differentiable training manner. Thus, we perform experiments by warming up the codebooks by *Initialization* and we get the following results in terms of Recall@10 on four datasets as Fig. 2. We adopt the early stop strategy to get the best model.

**Initialization.** We obtain the pre-trained codebooks by iterative training manners and continue differentiable training when the index assignment is approaching being balanced ($max|\mathcal{P}_i| - min|\mathcal{P}_j| < \frac{N}{W}, i, j \in W$) where $|\mathcal{P}_i|$ denotes the length of the $i$-th posting list. To accelerate the iterative training, codebooks are warmed by original quantization methods such as PQ, OPQ, and AQ.

**Findings.** KDindex converges to a better solution through the differential training manner. Within the dozens of epochs, the index assignment of iterative training becomes balanced, which warms up the centroids for later easier learning and thus relatively reduces the learning difficulty for both codebooks and indexes. Starting from this point, KDindex with differentiable training consecutively outperforms that with iterative training, which achieves a relative improvement of 1.63% in terms of Recall@10 on both datasets, demonstrating the effectiveness of synchronizing updates for codebooks and indexes. As for the different quantization functions, the improvements of Recall@10 among different student models (PQ, OPQ, and AQ) are 1.49%, 1.46%, and 1.94%, respectively. The better performance of KDindex(AQ) may be attributed to its better expressiveness with more parameters. Finally, the improvement of Recall@10 on MS MARCO Doc by KDindex(PQ) is 0.40%, which is smaller than the other model since the express ability of PQ is limited. The improvement of Recall@10 on SIFT1M by KDindex(AQ) is 0.39% since the express ability of AQ is strong on the L2 distance dataset and no more improvements can be obtained easily.

### 4.4 Memory Efficiency

As shown in Table 3, KDindex is more memory-efficient than the non-exhaustive ANNS method, HNSW, while performing comparable effectiveness. The index size of KDindex is less than 0.6GB, while HNSW requires storing all the original vectors and the graphs and consumes 0.5GB, 3.8GB, 9.2GB,

Table 3: Memory (GB) of Indexes on different ANNS methods. Compression Ratio denotes the ratio between the memory of the non-exhaustive method (HNSW) and KDindex.

| Index(GB) | SIFT1M | GIST1M | MS MARCO Doc | MS MARCO Passage |
|---|---|---|---|---|
| HNSW | 0.51200 | 3.84000 | 9.19480 | 25.2967 |
| KDindex | 0.07036 | 0.06064 | 0.19962 | 0.53507 |
| Compress Ratio | 7x | 63x | 46x | 46x |

and 25.3GB on four datasets respectively for storing indexes. The average Compression Ratio between the memory of the non-exhaustive method (HNSW) and KDindex on four datasets approximates 40x, demonstrating the superiority of the low memory cost of KDindex owing to the lightweight structure.

## 4.5 Hyperparameter Analysis

In this section, we analyze the effects of hyperparameters, including the number of nearest neighbors retrieved from the teacher model $K$, the number of codebooks $B$, and the number of codewords in each codebook $W$.

**Effects of $B$ and $W$** Figure 3 shows the effects of $B$ and $W$, where $B$ varies in $\{4, 8, 16, 32\}$ and $W$ in $\{64, 128, 256, 512\}$. We conduct experiments on SIFT1M and compare the results of KDindex with different quantization methods.

Firstly, we see that, in terms of task performance, decreasing $W$ has a far more traumatizing impact on KDindex. This is due to the nearest neighbor estimate losing accuracy when the number of codewords in each codebook drops.

Secondly, increasing $B$ or $W$ would typically improve the task performance at the expense of lower efficiency and more storage cost, which means one can adjust $B$ and $W$ in the meantime to obtain a trade-off between effectiveness and efficiency.

Thirdly, we notice that the small $W$ and large $B$ combination is preferable to the opposite way around. For instance, among the three quantization methods with KDindex, ($B = 32$; $W = 64$) performs better than ($B = 4$; $W = 512$) on the SIFT1M.

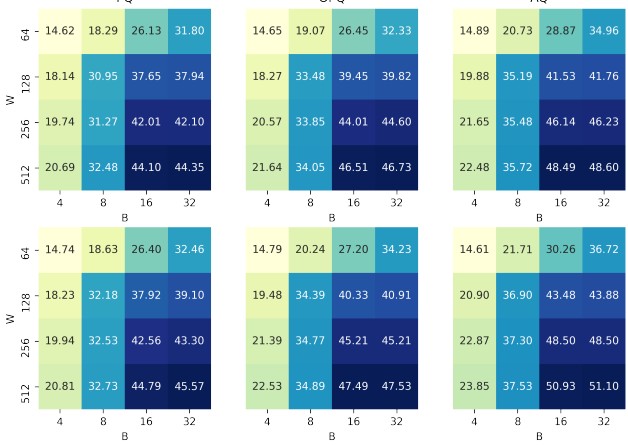

Figure 3: Heat-maps of performance for various $B$ and $W$ values on SIFT1M. The first line is the results of the original Quantization methods on Recall@10. The second line is KDindex based on corresponding quantization methods. Darker is better.

Table 4: The results under the different number of nearest neighbors $K$ in distillation loss on MS MARCO Doc.

| | $K$=5 | $K$=10 | $K$=20 | $K$=50 | $K$=100 |
|---|---|---|---|---|---|
| Recall@10 | 18.32 | 18.93 | 17.56 | 17.18 | 16.43 |
| MRR@10 | 41.32 | 41.69 | 40.33 | 39.44 | 39.03 |

**Effects of $K$** The relevant results come from the teacher model directly influence the distribution between the teacher and student and thus play a vital role in distillation loss. We vary the number of the nearest neighbors among $\{5, 10, 20, 50, 100\}$ and report the results in Table 4.

The model performs best when the number of neighbors is equal to 10, which is consistent with the cutoff. When the number of neighbors increases, leading to more data points are trained, the effects

degrade instead, which is contrary to the intuition that more candidates get better. The reason may lie in that the lower-ranked items have lower relevance with lower confidence and may bring more noise in training compared with the high-ranked items.

## 4.6 Ablation Study

KDindex includes three strategies, i.e., distillation loss, balance strategy, and differentiable training. We conduct an ablation study by incrementally adding the strategies to the basic quantization methods (PQ, OPQ, and AQ) to investigate their contributions. Specifically, we use the following model variants: (1) **Quantization** [24, 18, 3]: They are general quantization methods that serve as the backbone of the student model in KDindex. (2) **Initialization** is warmed up by quantization methods. It updates index assignments and centroids iteratively. is warmed up by quantization methods. It updates index assignments and centroids iteratively. (Details can be found in Section 4.4. We obtain the pre-trained codebooks by iterative training manners. ) (3) **w/o Distillation loss** denotes the training without knowledge distilled from the teacher model (HNSW in the experiment). It optimizes the centroids and trains the encoder under the constraint of reconstruction loss and balance strategy. (4) **w/o Balance strategy** denotes methods that without Sinkhorn-Knopp balance strategy. The reconstruction loss is so necessary to avoid a trivial solution that could not be removed. (5) **KDindex** denotes methods that differentially train models with Reconstruction loss, Distillation loss, and Balance strategy.

Table 5: Ablation study on MSMARCO Doc dataset.

| Model variants | | Quantization | Initialization | w/o Distillation loss | w/o Balance strategy | KDindex |
|---|---|---|---|---|---|---|
| PQ | Recall@10 | 2.85 | 6.32 | 6.69 | 8.62 | 8.74 |
| | MRR@10 | 3.82 | 14.33 | 15.27 | 17.01 | 17.64 |
| OPQ | Recall@10 | 14.63 | 15.87 | 15.93 | 16.25 | 16.70 |
| | MRR@10 | 34.75 | 38.96 | 39.01 | 39.22 | 39.75 |
| AQ | Recall@10 | 16.73 | 17.39 | 17.60 | 18.26 | 18.93 |
| | MRR@10 | 38.52 | 36.52 | 37.02 | 40.39 | 41.69 |

We conduct experiments on the MS MARCO Doc dataset with the same settings as mentioned above and report MRR@10 and Recall@10. Results are shown in Tab. 5. We can see that all three strategies contribute to the effectiveness of KDindex. With the help of the distillation loss, Recall@10 and MRR@10 achieve 14.34% and 10.01% relative improvements respectively. This indicates that the knowledge distilled based on the ranking list retrieved from the teacher model contributes to the learning of the centroids in the student model. Furthermore, the posting list balance strategy leads to a relative 2.61% and 2.76% improvement on Recall@10 and MRR@10, demonstrating that the avoidance of trivial solutions by controlling the length of the posting list is beneficial to building more appropriate centroids.

## 5 Conclusion

This paper presents KDindex, a novel framework for learning compressed indexes in high dimensions, which distills knowledge of the top-k relevant documents from the teacher index into the lightweight indexes to improve the retrieval accuracy for the student search model. To guide the student model to learn the same ranking list, rank-oriented distillation losses are designed and the posting list balance strategy is proposed to avoid the trivial solutions implemented with the Sinkhorn-Knopp algorithm. The learning of the student search model consists of the centroids embedding learning and the index assignment is exploited to realize the end-to-end differentiable learning. We conduct experiments on four publicly available benchmarks, where KDindex achieves impressive performance. Even if KDindex utilizes the approximated nearest neighbors and compresses the index by 40x, it outperforms some competitive models regarding ground truth generated by bruce-force as labels and gains better ranking performance than the state-of-the-art one.

## Acknowledgments and Disclosure of Funding

The work was supported by grants from the National Natural Science Foundation of China (No.62022077 and 61976198).

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

# A  Related Work

## A.1  ANNs and MIPS

ANNS achieves highly efficient vector search by allowing a small number of errors. Generally, there are two kinds of ANNS algorithms: *non-exhaustive ANNS methods* [39, 36, 38, 29] and *vector compression methods* [12, 3, 20]. Specifically, *Non-exhaustive ANNS methods* do not compress the index. They reduce the number of candidates for each query to speed up the retrieval process. Popular algorithms include tree search [16, 15] and graph search [36, 38, 29]. *Vector compression methods* mainly aim to compress the index to accelerate retrieval. Popular algorithms include hashing [12, 40, 43] and quantization [3, 20, 19].

Under the constraints of storage, compressed methods are widely investigated by researchers. Product quantization [24, 7] decomposes the vector representation space into the Cartesian product of subspaces. Optimized product quantization (OPQ) [18] jointly learns space decomposition and subspace quantization. Multi-scale Quantization [50] includes a multi-scale framework so that it can learn a separate scalar quantizer. Composite Quantization [54] and Additive Quantization [3] do not decompose space but directly learn multiple codebooks. There are also some algorithms that take query information into account. NEQ [11] decomposes the quantization error into norm error and direction error and improves existing VQ techniques for MIPS. ScaNN [20] computes the weight for each pair of vectors. Different from NEQ and ScaNN, KDindex utilizes query and corresponding top-k candidates. LSH (Local sensitive hashing) is a data-independent un-supervised method, similar to those clustering-based conventional quantization methods. BLISS [21] regards ground truth as labels. However, the ground truth is difficult to obtain in huge quantities of databases. LSH approaches have the property that objects that are close to each other have a higher probability of colliding than objects that are far apart across various distance metrics. The drawbacks of these approaches are the requirement for a large number of hash tables in order to achieve good search quality and these methods are unmindful of the distribution of vectors, often leading to lop-sided partitions and long query times. Interested readers could refer to the surveys [46, 31].

## A.2  Knowledge Distillation

*Knowledge Distilling* (KD) was first proposed in [22], in which a complex neural network was firstly trained and then transferred to a small model. Following this, DarkRank [8] proposed a method combining deep metric learning and *Learning to rank* technique with KD to solve image retrieval and image clustering tasks. In addition, a few recent methods [30, 41] have adopted knowledge distillation to RS. RD [44] firstly proposes a KD method that makes the student give high scores on the top-ranked items of the teacher's recommendation list. Similarly, CD [30] makes the student imitate the teacher's prediction scores with particular emphasis on the items ranked highly by the teacher. The most recent work, RRD [27], formulates the distillation process as a relaxed ranking matching problem between the ranking list of the teacher and that of the student. However, there are limited works focusing on index building under knowledge distillation.

There are also numerous works that use knowledge distillation to improve the performance of hashing-based [45] and quantization-based [51] codes, The ranking information is distilled from a graph-based network to enhance the performance of hashing codes. However, these works rely on the ground-truth labels (user-item interactions) to learn the ranking orders. This is different from our work, where label information is not accessible for learning. The quantization based method uses knowledge distillation for ranking candidates in web search tasks and applies the sampling technique to rank a sample of the document from data each time. But this technique is not applicable to training a ranking model when documents and queries are represented with no content information. In this case, the labeled model training cannot be easily generalized to all documents and queries. In contrast, KDindex relaxes the requirement on labeled data and can be trained purely with unlabeled data.

Table 6: Statistics of the datasets.

| Datasets | #Database | #Train | #Test | Dim |
|---|---|---|---|---|
| SIFT1M | 1,000,000 | 100,000 | 10,000 | 128 |
| GIST1M | 1,000,000 | 500,000 | 1,000 | 960 |
| MS MARCO Doc | 3,213,833 | 367,013 | 5,193 | 768 |
| MS MARCO Passage | 8,841,823 | 808,731 | 101,093 | 768 |

## B More Details of Experimental Settings

### B.1 Dataset Statistics

### B.2 Baselines

The two groups of baseline ANNS models are compared to KDindex.

The first group is *Non-quantization-based ANNS methods*, which accelerate the search by reducing the number of candidates. **BLISS** [21] adopts the learning-to-index framework to learn the hashing-based compressed functions. **ScaNN** [20] quantizes with anisotropic quantization loss functions which greatly penalizes the parallel component of a datapoint's residual relative to its orthogonal component. **HNSW** [37] builds a hierarchical set of proximity graphs. Results can be found in Appendix D.

The second is *Quantization-based ANNS methods*, which compress the embeddings by hashing or quantization functions. **PQ** [24] decomposes the vector representation space into the Cartesian product of subspaces. **OPQ** [18] jointly learns space decomposition and subspace quantization. **AQ** [3] represents each vector as a sum of several components each coming from a separate codebook. The baselines are implemented based on the Faiss ANNS library [26]. The parameters $B$ and $W$ are set to be the same as KDindex. **DiffPQ** [6], differentiable product quantization, a generic and end-to-end learnable compression framework. **DeepPQ** [17], deep progressive quantization, end-to-end learns the quantization codes sequentially. **PQ-VAE** [49], an unsupervised model for learning discrete representations by combining product quantization and auto-encoders. The CNN blocks are replaced with MLP because the image datasets have been extracted as 512-dimension features. **GCD** [25] learns rotation matrix via a family of block Givens coordinate descent algorithms. **RepCONC** [53] requires data points to be uniformly clustered around the quantization centroids.

### B.3 Implemental details

Table 7: Details of teacher model (HNSW).

| Teacher (HNSW) | SIFT1M | GIST1M | MS MARCO Doc | MS MARCO Passage |
|---|---|---|---|---|
| M | 32 | 32 | 32 | 32 |
| efConstruction | 40 | 40 | 100 | 100 |
| efSearch | 100 | 512 | 1024 | 1024 |
| Search Time (s) | 0.5862 | 1.3082 | 1.4805 | 4.7689 |
| Building Time (s) | 20.1s | 2m25.4s | 17m52s | 98m17.2s |
| Recall@10 | 0.9865 | 0.9859 | 0.9292 | 0.9182 |

Teacher model is instantiated as HNSW. The details are described as Tab. 7, where M denotes the number of neighbors each node, efConstruction denotes expansion factor at construction time and efSearch denotes expansion factor at search time. To obtain good recall performance, M, efConstruction and efSearch are tuned. Building and Searching are on multi-threads for train sets.

## C Varying Distillation Loss

### C.1 Distillation Losses

Knowledge distillation was first proposed for classification tasks, where the probabilities of each class attained from the large-scale teacher network are considered as soft labels to supervise the learning

of the small-size student network. The cross-entropy loss is commonly used as the distillation loss to minimize the difference between the teacher and student networks. Here, the teacher search model provides the top-k relevant candidates rather than the continuous value of probabilities. Thus, three ranking-oriented losses are designed to distill knowledge from the more accurate indexes to guide the student indexes to return the same nearest results.

**Lambdarank loss**: The pair-wise ranking-based loss is widely used to learn the ranking list by leading the high-ranked candidate to have higher similarity scores. Lambdarank [4] further introduces the change of the indicators, e.g., NDCG, to put more attention on more important candidates that have not been well ranked. The loss follows as:

$$\mathcal{L}(\boldsymbol{q}, \mathcal{D}_K^T; \boldsymbol{C}) = \sum_{i,j \in \mathcal{D}_K^T} \log\left(1 + \exp(p_i - p_j)\right) |\Delta NDCG@10_{ij}| \tag{4}$$

where $\mathcal{D}_K^T$ denotes the top-k results retrieved from the teacher model and $p_i = S(\boldsymbol{q}, Q(\boldsymbol{d}_i))$ is the similarity score between the query vector and the quantized vector of the candidate $i$. $Q$ is the quantizer function related to the codebooks $\boldsymbol{C}$. $\Delta NDCG@10_{ij}$ denotes the change with respect to NDCG@10 if changing the $i$-th ranked and $j$-th ranked candidate.

**Weighted KL loss**: Similar to the class distribution in classification tasks, the similarity distribution over the top-k retrieved candidates can also be obtained. One is based on the ground-truth vectors and the other one is based on the quantized vectors. In order to ensure the ranking orders correspond to the top-k list, the rank information is also considered where the high-ranked items are more concerned. Finally, the loss function follows as:

$$\mathcal{L}(\boldsymbol{q}, \mathcal{D}_K^T; \boldsymbol{C}) = - \sum_{i \in \mathcal{D}_K^T} \tilde{p}_i^g \log \frac{\tilde{p}_i^g}{\tilde{p}_i} \tag{5}$$

where $\tilde{p}_i$ denotes the normalized ranked similarity score with the quantized vector and $\tilde{p}_i^g$ with the ground-truth vector. Specifically,

$$p_i = w_i \cdot S\left(\boldsymbol{q}, Q(\boldsymbol{d}_i)\right), \quad p_i^g = w_i \cdot S(\boldsymbol{q}, \boldsymbol{d}_i),$$

$\tilde{p}_i$ and $\tilde{p}_i^g$ are the normalized values over the top-k retrieved candidates depending on the softmax function. $w_i = \frac{1}{rank(i)}$ denotes the ranking weight according to the ranking orders among the top-k results from the teacher model. The weighted KL loss attempts to minimize the distance between the ground-truth vector and the quantized vector for the top-k relevant candidates to learn better centroids. The introduced rank-oriented weight further guides the student index to return the same ranking list.

**Distributed-based loss**: Instead of being oriented by the score between query and candidates as above, we attempt to minimize the distance between the queries and top-k neighbors by calculating the similarity scores with all the centroids. Thus, we could obtain more information from centroids and focus on the top-K nearest neighbors. The distributed-based loss function follows as:

$$\mathcal{L}(\boldsymbol{q}, \mathcal{D}_K^T; \boldsymbol{C}) = - \sum_{i \in \mathcal{D}_K^T} \sum_{b=1}^{B} \sum_{k=1}^{W} \tilde{p}_{bk}^q \log(\tilde{p}_{bk}^{d_i} \cdot w_i) \tag{6}$$

where $B$ denotes the number of codebooks and $W$ is the number of codewords in each codebook. $w_i = \frac{1}{rank(i)}$ corresponds to the top-k list given from the teacher model. $p_{bk}^q$ denotes the similarity score between the query $\boldsymbol{q}$ and the codeword $\boldsymbol{c}_k^b$, i.e., $p_{bk}^q = S(\boldsymbol{q}, \boldsymbol{c}_k^b)$, and $p_{bk}^{d_i}$ denotes the similarity score between the candidate $\boldsymbol{d}_i$ and the codeword $\boldsymbol{c}_k^b$, i.e., $p_{bk}^{d_i} = S(\boldsymbol{d}_i, \boldsymbol{c}_k^b)$. The normalized value $\tilde{p}_{bk}^q$ and $\tilde{p}_{bk}^{d_i}$ are calculated over the $W$ codewords for each codebook through the softmax function.

This loss requires the enumeration of all the centroids, while the Weighted KL loss only includes parts of the centroids corresponding to the quantized function. It also aligns with the goal of nearest searching for the query with the learnable centroids as the bridge.

## C.2 Experimental Performances

We compare the effectiveness of the three different distillation losses, i.e., Weighted KL loss, Distributed-based loss, and Lambdarank loss as reported in Table 8.

**Findings.** Overall, the Distributed-based loss leads to comparatively better performances than Weighted KL loss and Lambdarank loss. Compared with Weighted KL Loss and Lambdarank Loss, Distributed-based Loss gains the 2.03% and 3.54% improvements of Recall@10, 0.38%, and 0.70% of NDCG@10, respectively. The Lambdarank Loss concerns more about the relationships between the pair of items, while the other two care about the whole ranking order of the list. The weighted KL loss actually optimizes parts of the centroids, depending on which query vectors and candidate vectors are quantized, to match the ranking list, while all of the centroids are updated in the Distributed-based loss since the probabilities are calculated over all the centroids. Furthermore, the Distributed-based loss requires the similarity calculation between the original input vectors and the centroids, which eliminates the error caused by the compressed functions. The last observation is that Distributed-based Loss works better on inner product metric datasets, since it obtains the average improvements of 2.24% and 3.33% of Recall@10 on L2 distance and inner product, respectively, wherein the overall improvements for inner-product similarities.

Table 8: The results of KDindex under different distillation loss functions.

| Loss Function | SIFT1M | | GIST1M | | MS MARCO Doc | | MS MARCO Passage | |
|---|---|---|---|---|---|---|---|---|
| | Recall@10 | NDCG@10 | Recall@10 | NDCG@10 | Recall@10 | MRR@10 | Recall@10 | MRR@10 |
| Lambdarank Loss | 36.32 | 79.18 | 20.69 | 62.06 | 17.60 | 40.01 | 11.10 | 34.86 |
| Weighted KL Loss | 36.68 | 79.33 | 21.02 | 62.75 | 18.24 | 40.93 | 11.06 | 34.63 |
| Distributed-based Loss | 37.30 | 80.01 | 21.33 | 63.17 | 18.93 | 41.69 | 11.19 | 35.23 |

# D  Search Efficiency and Retrieval Quality

The search efficiency and the comprise performance are important indicators of a compression search index. Thus, we report the effectiveness-latency performance for inference in Figure 4.

The time efficiency is measured by the latency of getting the Top-10 candidates for each query. The retrieval performance and the running time are governed by the number of posting lists we probe among the codebooks. The compressed ANNS methods (ScaNN and BLISS) and an uncompressed method (HNSW) are introduced for comparison. We can see that KDindex substantially outper-

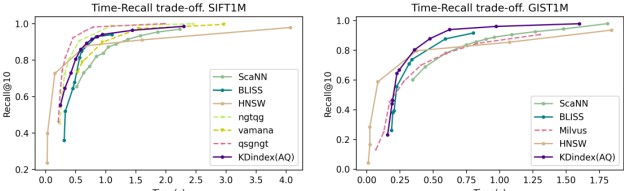

Figure 4: Comparison with more compressed ANNS models and uncompressed method (HNSW, ngtqg, qsgngt, vamana, and Milvus) in terms of effectiveness-latency trade-off on two datasets. Up and left is better. The results of ScaNN, HNSW, ngtqg, qsgngt, vamana, and Milvus come from ANN-Benchmarks [1].

forms baselines in terms of both effectiveness and time efficiency. Specifically, we find that KDindex cost less time to arrive at a similar Recall than both ScaNN and BLISS. Although HNSW can achieve better results in the early stage, KDindex will surpass it after half a second.

# E  Limitations and Future Works

In this paper, we propose a novel knowledge distillation framework for high dimension index, which reduce storage obviously and can learn neighbor information from the teacher model. Especially, KDindex is independent to label (such as interaction information in the recommendation system or ground-truth neighbors in ANNS), which makes it flexible to be applied in more label-free scenarios. In the future, we will try more student models such as lattice quantization, whose codes already imply neighbors relationship. And we will take labels into account to improve retrieval performance progressively. We will further improve our work to benefit the broad community.

