# OpenReview forum: "Knowledge Distillation for High Dimensional Search Index"
_NeurIPS.cc/2023/Conference — NeurIPS 2023 poster_

### Official Review · Reviewer_SPup · 2023-07-05

**Soundness:** 3 good
**Presentation:** 3 good
**Contribution:** 2 fair
**Rating:** 5
**Confidence:** 5

**Summary:**

The paper proposes a new method called KDindex to learn lightweight indexes by distilling knowledge from high-quality ANNS. The method outperforms existing learnable quantization-based indexes and state-of-the-art ANNS methods by learning to keep the ranking order, adding the reconstruction loss to minimize the compression error, and adopting a balanced partition strategy

**Strengths:**

KDindex achieves good performance which outperforms existing learnable quantization-based indexes and some well-performed ANNS methods.

**Weaknesses:**

1.	This paper misses some related work, such as Poeem, JPQ, and MoPQ.
2.	The idea of this paper is similar with Distill-VQ. It improves the Distill-VQ by adding reconstruction loss into the learning objective and adjusts the posting list according to a balance strategy. I would like to see the comparison with Distill-VQ.
3.	For search efficiency and retrieval quality experiment, it does not compare with the state-of-the-art ANN algorithms in the ANN-Benchmarks, such as NGT-qg, qsgngt, and vamana.
4.	It seems balance strategy does not help too much.


**Questions:**

Please address the above weaknesses.

**Limitations:**

It does not include limitation analysis. Please add a Limitations section.

---

> ### Author Rebuttal · Authors · 2023-08-10
>
> 1.  [Other related works]
>
>     > **Comments:** This paper misses some related work, such as Poeem, JPQ, and MoPQ.
>
>     Thank you for your valuable advice. The mentioned related works, including Poeem, JPQ and MoPQ, focus more on the joint learning of both retrieval embedding models and the quantization models, where their aim is to integrate the separate encoding and compression process into an end-to-end training process.
>
>     Different from these works, our work mainly focuses on the better compressed methods given well-trained embedding vectors, where we
> do not focus on the encoding models. We will elaborate on this difference in our work.
>
> 2. [Discussion with Distill-VQ]
>
>     > **Comment:** The idea of this paper is similar with Distill-VQ. It improves the Distill-VQ by adding reconstruction loss into the learning objective and adjusts the posting list according to a balance strategy. I would like to see the comparison with Distill-VQ.
>
>
>     Indeed, Distill-VQ incorporates the distillation method to jointly learn the query encoder and the index models. Distill-VQ can relax the requirements of ground-truth labels while KDindex is designed for unlabeled data. However, there are several differences between Distill-VQ and KDindex.
>
>     The first distinction lies in the dense embedding encoders. Distill-VQ necessitates the learning of query encoders for documents or  images to align with the candidate document embeddings. For KDindex, all the high dimensional embeddings are fixed so that it is a pure ANNs problem without the attention for encoding queries since we have no content information.
>
>     The second differentiation revolves around the nature of distillation signals. Distill-VQ takes the similarity information of fixed document embeddings as signals by disclosing the scores between the original fixed embeddings and the reconstructed embeddings. However, KDindex aims to preserve the ranking information from a teacher index model, where the teacher index model directly provides the retrieved top-k results. Thus, the sampled candidate document set plays an important role in Distill-VQ, which just represents local similarity information in the case of a huge document size.
>
>     For better compare the performance of different distillation methods, we conduct experiments for better illustration. For fair comparison, all the embeddings are fixed in Distill-VQ and the index structure follows AQ.  We adopt the in-batch sampling method with a batch size of 64 in Distill-VQ and calculate the distillation loss with  KL-divergence loss function. The following table shows the results:
>
>     | Models       | SIFT1M    |         | MS MARCO Doc |        |
>     | ------------ | --------- | ------- | ------------ | ------ |
>     |              | Recall@10 | NDCG@10 | Recall@10    | MRR@10 |
>     | Distill-VQ   | 35.79     | 78.43   | 17.34        | 39.93  |
>     | KDindex (AQ) | 37.30     | 80.01   | 18.93        | 41.69  |
>
>     As shown in the above table, KDindex still outperforms Distil-VQ, demonstrating the superiority of the distillation methods in KDindex.
>
> 3. [Graph-based ANN Benchmarks]
>
>     > **Comments:** For search efficiency and retrieval quality experiment, it does not compare with the state-of-the-art ANN algorithms in the ANN-Benchmarks, such as NGT-qg, qsgngt, and vamana.
>
>
>     The results of NGT-qg, qsgngt, Milvus and vamana are added in Fig 2, which can be referred in Figure 6 in the supplement PDF. These alogrithms are graph based and full percision vectors are stored on node of graph. These graph-based methods show better performances especially with more latency. All these graph-based method can act as teacher model to train our KDindex.
>     We point that our KDindex is designed to enhance the performance of compressed index with the help of better-performed teacher index rather than defeat those graph-based methods.  As for compressed methods, such as ScaNN and BLISS, KDindex can perform better. In the futrue work, we will try more efficient method as teacher index to further improve compressed index.
>
>
>
> 4. [Balance Strategy]
>
>     > **Comments:**  It seems balance strategy does not help too much.
>
>      Compared with other important parts, the relative 2.61% and 2.76% improvements in Recall@10 and MRR@10 of posting balance strategy are smaller but it still accounts for significant retrieval improvements.

---

> > ### Comment · Reviewer_SPup · 2023-08-22
> >
> > Thank you for the rebuttal! For Q2, Distill-VQ can also fix query encoder to just learn the quantization codebook. It is also optimized to perserve the ranking order of topk results. It uses the ListNet loss instead of the KL-divergence loss. Could you provide the results of Distill-VQ with the right setting? For Q3, where can I find the results? I cannot see the results in supplement PDF. You can distill KDindex from the graph-based ANN methods. However, how can you outperform these indices when application scenarios require high recall since quantization is a lossy compression solution?

---

### Official Review · Reviewer_aAUh · 2023-07-05

**Soundness:** 4 excellent
**Presentation:** 2 fair
**Contribution:** 3 good
**Rating:** 7
**Confidence:** 4

**Summary:**

This paper proposes to train an quantization-based approximate nearest neighbor search index using query and its approximate kNN items which are obtained using a teacher kNN search model. Authors propose loss functions to incorporate the ranking of kNN items as well as additional constraints to prevent trivial solutions and improve quality of approximation. The proposed framework yield significant improvement over baseline quantization methods as well as previous work on learning-based  quantization approaches.

Overall score updated from `6: Weak Accept` to `7: Accept` after author response.

**Strengths:**

- The proposed method yields strong empirical results and experiment section includes ablation study, analysis of sensitivity to hyper-parameters, and comparison with state-of-the-art methods.
- Each component of the proposed work is well-motivated and is accompanied with corresponding ablation results (although some of them have been moved to appendix but I would encourage authors to include some more details about baselines in the main paper).

**Weaknesses:**

- The presentation of the method and results can be improved further.
    - For instance, a clear description of test-time inference process would be helpful.
    - Also, it is not intuitively clear why using query-item interactions can results in better quantization for kNN search. The goal of the quantizer is to accurately express the original distance function. Does the proposed method improves kNN search performance at the expense of quantization performance or is the resulting quantization also a better approximation of the original distance function.
- Missing information on Indexing time.
    - While the proposed method does improve test-time performance, it would be interesting to see how much time does it take to index a given set of items
- (Minor) Distillation is perhaps not the right term here.
    - The model is trained using approximate nearest neighbors as per an expressive kNN search index.  But unlike “standard” distillation papers, the teacher model is not used to provide any soft training signal. The only training signal from query-item interactions is the ranking of items and this ranking is not induced by the teacher model but by the underlying similarity function.
    - Why did the authors not used exact kNN items to train the index? I understand that obtaining exact kNN item might be significantly slower than retrieving approximate kNN items using teacher kNN index. But finding exact kNN items is a one-time cost and may turn out to be a small fraction of the overall running/training time as this exact kNN may not be the as expensive as the subsequent training of the kNN index.
- (Minor) Proposed method requires a set of training queries to index the items. In typical settings for kNN search, the indexing algorithm has access to only the set of items. It would be interesting to use a subset of items as (pseudo-)queries in order to learn such an index.

**Questions:**

- What is the accuracy of the teacher model? Consider adding performance of teacher model in Table 1.
- What is the intuition behind query-item interaction being helpful over just item-based quantizers?
- Eq 6 is references in Algo 2 but I could not find it in the paper. Did authors mean eq 1?
- Description of `w/o distillation` ablation says that “trains the encoder with knowledge distilled from the teacher model (HNSW in the experiment).” Can authors elaborate on this point in line 229? “*The improvement of KDindex is more significant when the distribution between query space and database space is different”.*
    - What do authors mean in Line 231 by “The similarity could not  be obtained by original quantization methods”? Is this some fundamental limitation of the quantization-based methods?
- Is it not clear how the proposed method avoid alternating between updating codewords and query/item-to-codeword assignments.
    - In line 14 as Algo 2, as the codewords are updated, the index assignments in computed in Line 5 will become out-dated, right?
    - Are the index assignments (assignments of documents to codewords) also updated somehow together with the codewords in line 14?

---

> ### Author Rebuttal · Authors · 2023-08-10
>
> 1.  [test-time inference process]
>
>      > **Comment**: a clear description of test-time inference process would be helpful.
>
>     For comparsion among quantization methods in Table 1,  Asymmetric distance computation (ADC) is used.
>     To compare with more ANNS methods in Table 2, an inverted file system is combined with the asymmetric distance computation (IVFADC). This allows rapid access to a small fraction of database indices and is shown to be successful for very large scale search. More detail can be found in [1].
>     For KDindex, Table 1 is based on ADC and Figure 2 is based on IVFADC.
>     [1] Product quantization for nearest neighbor search. TPAMI 2020.
>
> 2. [Motivation of  query-item interactions]
>
>     > **Comment**: why using query-item interactions can results in better quantization for kNN search.
>
>     According to previous work[1], the introduction of query-item interactions would increase the probability of similar items assigning to the same centroids. Similarly, we extend the theoretical results to KDindex, which adopts the quantization methods rather than hashing based methods in [1].
>     For the training query $q$ and its corresponding ground truth $p$ in the database, the expected probability of the centroids containing $p$ given the query $q$ increase by a positive margin after reassignment, i.e. $\mathbb{E}[𝑓^′(e^′(𝒑)|𝒒)] ≥ \mathbb{E}[𝑓 (e (𝒑)|𝒒)]$,where $f(\cdot)$means the scoring function given by the model and $e(\cdot)$means the quantization function that maps the points to centroids. The increment in this probability results in an increment in the quality of the retrieved candidates during inference.
>     The theorem implies that the centroids containing the relevant points derive a higher aggregated probability as it will contain other ground truths with higher probability. Consequently, query-item interaction is advantageous over item-based quantizers alone. The increment from interaction will be higher in the beginning as the relevant documents are quantized into the same centroids. As training goes on, the increment dies down to 0 and the model converges to the optimal quantization.
>
>     [1] BLISS: A Billion scale Index using Iterative Re-partitioning. SIGKDD 2022
>
> 3. [Missing information on Indexing time.]
>
>     In the indexing stage, the codebooks are well learned and the indexing time of KDindex is same as corresponding basic quantization. We detail the time complexity in global response PDF.
>
> 4.  [Distillation Details]
>
>     > **Comments:** Distillation is perhaps not the right term here
>
>     Conventional knowledge distillation fits in the classification tasks, where the more powerful teacher model imparts the soft labels to guide the learning of the student models. However, in the ranking tasks, such as recommendation and retrieval tasks, ranking distill methods are proposed, such as RD[1], RankDistill[2], to cater to the unique demands of ranking-oriented tasks. The ranking knowledge is distilled to the student model to learn a more accurate ranking performance. In this paper, KDindex adopts a more direct form of ranking signals from teacher index, where the approximate nearest neighbours are retrieved by the teacher index and the student models aim to preserve the ranking performance with the quantized embeddings. These retrieved topk items are different from different training queries and thus enhance the ability to capture similarities and conduct effective searches.
>     [1] Ranking distillation: Learning compact ranking models with high performance for recommender system, KDD 2020
>     [2] Rankdistil: Knowledge distillation for ranking, ICML 2021
>
>
> 5.  [Pseudo Query for training]
>
>     The large scale datasets with high dimensional vectors for ANN search usually include the training query vectors for learning and the test query vectors to fit in the dense retrieval scenarios.
>     For better illustrate the generalization ability, we conduct experiments with pseudo queries where we select 100K and 367K items as query vectors from the SIFT1M and MS MARCO Doc datasets. The results on SIFT1M of Pseudo query are approaching to the performance of True Query while on MS MARCO Doc is worse. The reason lies in the significantly different distribution between the query and database, as shown in Figure 5 of the supplemental PDF. Thus, distillation information from query vectors is helpful to learn codebooks.
>
>     |KDindex (AQ) | SIFT1M  |           |         | MS MARCO Doc |           |        |
>     | ------------ | ------- | --------- | ------- | ------------ | --------- | ------ |
>     |              | #query  | Recall@10 | NDCG@10 | #query       | Recall@10 | MRR@10 |
>     | True query   | 100,000 | 37.30     | 80.01   | 367,013      | 18.93     | 41.69  |
>     | Pseudo query | 100,000 | 37.15     | 79.43   | 367,013      | 16.64     | 38.25  |
>
> 6. [Performance of Teacher Model]
>
>     For Table 1, we perform ADC over these quantization-based methods. Since HNSW is a graph-based method, the comparison with these quantization-based methods is not fair. We provide more details about training.
>
>     For training, We obtain the approximate top-K neighbors from teacher model and the performance is approaching ground-truth by adding the search latency. The time and performance used in experimental are as following: (More details in Appendix  B.3)
>
>
>     | Datasets       | SIFT1M | GIST1M | MS MARCO Doc | MS MARCO Passage |
>     | -------------- | ------ | ------ | ------------ | ---------------- |
>     | Recall@10      | 0.9865 | 0.9859 | 0.9292       | 0.9182           |
>     | NDCG@10        | 0.9999 | 0.9999 | N/A          | N/A              |
>     | MRR@10         | N/A    | N/A    | 0.9493       | 0.9327           |
>     | Query time (s) | 0.5862 | 1.3082 | 1.4805       | 4.7689           |
>
>
>     For more comparison about the recall-time performance, please refer to Figure 6 in the global response PDF.

---

> > ### Comment · Reviewer_aAUh · 2023-08-16
> > **Follow-up after author response**
> >
> > I would like to thank the authors for the clarifications provided.
> >
> > 1. Indexing Time Follow-up
> > I saw that the authors added indexing complexity details in the global author response pdf. While it is useful to understand asymptotic time complexity of these methods, the actual time taken for indexing might be more meaningful as the constants involved in training might be crucial. Specifically for each quantization method, I am curious about how much time does vanilla quantization method take and how much time does the corresponding KDIndex variant take.
> > Also, how can training time be independent of the total number of items ($N$)?
> >
> > 2. Re: Advantage of using query-item interactions?
> > Do you think using query-item interactions help in converging to better quantization parameters while vanilla quantization methods might converge to sub-optimal parameters? In this case, by better I mean wrt pure quantization loss. It would be interesting to see quantization based loss both vanilla quantization and corresponding KDIndex model to understand if the performance gain in k-NN search metrics for KDIndex is coming at the cost of some drop in quantization performance.
> >
> > Could you also elaborate on some of the questions in my original review?

---

> > > ### Author Response · Authors · 2023-08-18
> > > **Further Response to Reviewer aAUh**
> > >
> > > Thanks for the reviewer's valuable reply!
> > >
> > > 1. Indexing Time Follow-up
> > > (1) **actual time taken for indexing:** The indexing complexity of KDindex is the same as the basic quantization method since the structure of codebooks and the way to quantize are the same. In the inference stage, only codebooks (and rotation matrix in OPQ) are used. The neighbors in the distillation or balance strategy are unrelated to inference. The actual indexing time on SIFT1M (# items = 1M) is as follows:
> > > | indexing time (s) | basic quantization | KDindex |
> > > | ----------------- | ------------------ | ------- |
> > > | PQ          | 34.7339            | 34.9440 |
> > > | OPQ        | 40.2778            | 41.5282 |
> > > | AQ            | 59.8159            | 59.1787 |
> > >
> > >     (2) **independent of $N$:** In the training process, codebooks are not learned by using all items so $N$ is not included in the expression. The actual number of items used in training is related to training queries and their neighbors. For $M$ training queries of each batch, a fixed number $K$(constant) of neighbors for each query is used. We haven't put the constant term in the expression.
> > >
> > > 2. Pure quantization loss
> > >
> > > Using query-item interactions does help in converging to better quantization parameters. The pure quantization loss on SIFT1M is as follows. More important, query-item interactions provide neighbor information which is beneficial to divide data points (keep the same code for similar points and use different codes for points with large differences). The neighbor relationship is the internal cause while reconstruction loss is appearance.
> > > $L = \|x-Q(x)\|^2=\sum_d^D(x_d-Q(x)_d)^2$, where $D$ denotes dimension, $x$ denotes item vector and $Q(x)$ denotes quantized item vector.
> > >
> > > | pure quantization loss for each item $L$ | basic quantization | KDindex     |
> > > | --------------------------------------------- | ------------------ | ----------- |
> > > | PQ             | 23521.02141  | 23190.76753 |
> > > | OPQ                     | 21728.97262        | 20653.29408 |
> > > | AQ                                            | 19675.23785        | 19029.35791 |
> > >
> > > Due to the word limitation, we deleted some of the original answers, and now we add them as follows.
> > >
> > > 3.  [Misleading of Eq 6]
> > > Eq (6) is represented in Appendix, which actually is the same as Eq (1). We are sorry for this repeated reference.
> > >
> > > 4. [Details about experiments]
> > >
> > > > **Comment:** Description of w/o distillation ablation says “trains the encoder with knowledge distilled from the teacher model (HNSW in the experiment).”
> > >
> > > Thank you for your careful review and we are sorry for the misleading descriptions. We will modify it as follows:
> > > (1) Quantization.
> > > (2) Initialization is warmed up by quantization methods. It updates index assignments and centroids iteratively. (Details can be found in Appendix D. We obtain the pre-trained codebooks by iterative training manners. )
> > > Below three experiments are based on (2)Initialization in the differentiable training manner.
> > > (3) w/o Distillation loss denotes the training without knowledge distilled from the teacher model (HNSW in the experiment). It optimizes the centroids and trains the encoder under the constraint of reconstruction loss and balance strategy.
> > > (4) w/o Balance strategy denotes methods without Sinkhorn-Knopp balance strategy.
> > > (5) KDindex denotes methods that differentially train models with Reconstruction loss, Distillation loss, and Balance strategy.
> > >
> > > > **Comment:** Can authors elaborate on this point in line 229?“The improvement of KDindex is more significant when the distribution between query space and database space is different”.
> > >
> > > When the distribution of query vectors and database vectors are different, the query information is more important to centroid learning. Knowledge distillation plays an important role in distilling query information from the teacher index to the student index. Thus, the more important the query information is, the better distillation works well. The distribution of four datasets is attached in PDF.
> > >
> > > > **Comment:** What do authors mean in Line 231 by “The similarity could not be obtained by original quantization methods”? Is this some fundamental limitation of the quantization-based methods?
> > >
> > > Original quantization methods only learn database distribution, which is the fundamental limitation of quantization-based methods (such as PQ, OPQ, and AQ). To utilize the information of query vectors, existing works (ScaNN and QUIP) sample part of query vectors. Instead, KDindex learns from the teacher index.
> > >
> > > 5. [Details about codeword update and index assignments]
> > >
> > > The general process includes the codebooks updating and index assignments within a mini-batch. For better illustration, we detail the process with respect to each query within the mini-batch and present it with for loops. Actually, the calculation is aggregated within the mini-batch and the updating of codebooks is performed for each batch, followed by the reassignment of the index.

---

> > > > ### Comment · Reviewer_aAUh · 2023-08-19
> > > >
> > > > Thanks for the response.
> > > > I would vote in favor of accepting this paper and I have updated my score accordingly from `6:Weak Accept` to `7:Accept`.
> > > >
> > > > I would encourage authors to include these additional results and analysis in the paper, perhaps in the appendix if it is difficult to make room in the main paper.

---

> > > > > ### Author Response · Authors · 2023-08-19
> > > > > **Reply to the reviewer aAUh**
> > > > >
> > > > > Thanks for your appreciation and reply! We are glad that our rebuttal resolved your questions. We will include the additional results and analysis in the final version.
> > > > >
> > > > > Thank you very much,
> > > > >
> > > > > Authors

---

### Official Review · Reviewer_uFQG · 2023-07-09

**Soundness:** 2 fair
**Presentation:** 3 good
**Contribution:** 2 fair
**Rating:** 5
**Confidence:** 3

**Summary:**

This work addresses the problem of learning a lightweight index for high-dimensional similarity search problems. Lightweight index is desirable in many applications which can't afford a high precision heavyweight index due to higher storage cost or computational constraints. Unlike past work on lightweight index learning, this work assumes the availability of a heavyweight index at training time and explores the possibility of learning of the lightweight index with the guidance of the heavyweight index. It proposes a knowledge distillation framework for learning a lightweight index when no label information is available. The key idea is to use the top-K nearest neighbor results of every training query for guiding the learning of the lightweight index. This enables the use of ranking-oriented loss in the training of the  lightweight index. The work employs two tricks to avoid trivial and imbalanced solutions - i) a reconstruction loss that minimizes the distance between the query/candidate and its codeword ii) balancing of posting list.  The proposed framework is applied on four datasets and experimental results are discussed.


For every query, a cross-entropy like loss is computed with respect the top-K approximate nearest neighbors returned by the heavyweight teacher search model. This loss downweighs the similarity score between the codeword and the candidate with the reciprocal of the candidate's rank.


**Strengths:**

Experimental results show that knowledge distillation from a heavyweight index improves the retrieval performance of the lightweight indexes.

Ablation study shows that two of the three strategies employed make a significant difference to retrieval performance of the lightweight indexes.


**Weaknesses:**

As the knowledge distillation process involves retrieving the top-K results for each query using the teacher search model and this is repeated in each iteration, the computational cost of distillation is high.

Datasets used in the experiments are small in size (< 10M documents).

No discussion of and comparison with LSH and learning to hash techniques.


**Questions:**

"Randomly initialize 133 the centroids (codewords) and assign indexes" --> is it possible to take the help of teacher index to initialize in a more informed manner?

In Figure 2, what is the unit of time in x-axis?

In Figure 2, KDIndex stands for which specific student model? Is it PQ or OPQ or AQ?

In Figure 2, HNSW is time efficient compared to KDIndex for low to moderately high recall scenarios. However, in high recall scenario, HNSW is significantly worse than KDIndex. How can this happen as HNSW is the teacher and KDIndex is the student?

In Figure 2, best recall is achieved by ScaNN though this comes at increased search latency. However, KDIndex plateaus off quickly and increased search latency doesn't seem to help.

The recall numbers of KDIndex in Figure 2 for both SIFT1M and GIST1M datasets don't match with the recall numbers reported in Table 1. In fact Recall @10 in Table 1 for KDIndex is less than 35 for SIFT1M and less than 22 for GIST1M whereas Figure 2 reports Recall @10 as high as 0.8. What explains this discrepancy?

Table 1 should report the results for the teacher search index HNSW to give an idea of the relative performance.

 In Section 4.4, it would be interesting to know the effect of B and W on storage and search latency.

On what criteria was B = 8  W = 256 was chosen as the optimal hyper-parameter setting? Was this the optimal setting for all the datasets?

In Table 2, it would be interesting to include K = 1 and 2. As K = 5 and K = 10 give very similar recall and MRR, it would be good to find out if smaller K also does similarly well.

In Section 4.5 and Table 2, what exactly is initialization warmed by quantization methods? Are you initializing the centroids with those for AQ/PQ/OPQ instead of random initialization?

Of the three strategies employed by KDIndex, Balance seems to give least improvement in retrieval performance going b y Table 3 (for instance, PQ Recall @10 8.64 vs 8.62). However, Balance adds significant complexity to the training algorithm. It would be good to report and compare time taken for training KDIndex and w/o Balance in Table 3 to get a better understanding of the tradeoff between incremental improvement in retrieval performance and training complexity.

In Table 4, KDIndex refers to which of KDIndex(AQ), KDIndex(PQ), KDIndex(OPQ)?

In Table 4, why is compression for SIFT1M is much lower than GIST1M and other datasets (7 vs 63)?

Why are the similarity functions for MS MARCO different from that for SIFT1M and GIST1M?

What is the additional time complexity of KDIndex relative to AQ, PQ and OPQ?

Why haven't 10M SIFT dataset and 5M SIFT dataset been used in the experiments as done by [38]?

References [27] and [28] are one and the same!

Figure 2: "trage off" should be "trade-off"

**Limitations:**

The submission doesn't discuss the limitations of its work and the potential negative societal impact of their work.

---

> ### Author Rebuttal · Authors · 2023-08-10
>
> 1.  [computational cost of distillation]
>
> > **Comments**: As the knowledge distillation process involves retrieving the top-K results for each query using the teacher search model and this is repeated in each iteration, the computational cost of distillation is high.
>
> Thanks for your thoughtful approach to KDindex training. Initially, top-K results are fetched for all training queries after teacher index (HNSW) training, ensuring efficient retrieval. This step occurs once in real implementation, with results cached for future use, forming precomputed label generation. When reviewing the Algorithm, notably line 17, "IO cost" primarily matters. Cached results from file reduce computational load.
> After obtaining the top-K retrieved results, the time complexity of distillation loss calculation is $O(BWK)$ for each query. Compared with the quantization time, which takes $O(BWD)$, the distillation cost is comparable and acceptable.
>
> 2.  [large datasets] Refer to reponse 3 for Reviewer LzAv.
>
> 3.  [discussion and comparison with LSH and learning to hash techniques]
>
> > **Comments**: No discussion of and comparison with LSH and learning to hash techniques.
>
> Thank you for your valuable suggestions. Actually, we have compared KDindex with the state-of-the-art hashing learning based method, i.e., BLISS [2], which preserves the ground-truth ranking orders to learn bucket partition for large-scale data. The experiments in section 4.3 demonstrate that our proposed KDindex outperforms BLISS in both retrieval performance and efficiency.
>
> For more discussion about the hashing-based methods, they can be integrated into the discussion with quantization-based methods in introduction. LSH (Local sensitive hashing) is a data-independent un-supervised method, similar to those clustering-based conventional quantization methods. LSH approaches have the property that objects that are close to each other have a higher probability of colliding than objects that are far apart across various distance metrics. The drawbacks of these approaches are the requirement for a large number of hash tables in order to achieve good search quality and these methods are unmindful of the distribution of vectors, often leading to lop-sided partitions and long query times.
>
> Learning to hash is also an effective method to compress high-dimensional data into low-dimensional binarized codes, where two types of loss functions are usually used. One is the reconstruction loss, which minimizes the distance between the original vector and the encoded vectors. The another one is the _ranking-based loss_, e.g., triplet loss and pair-wise loss, which encourages the model to learn a hash function that preserves the pairwise similarity relationships between positive points and negative points. However, only when the data set *contains the interaction information*, such as user clicks, the ranking loss would be utilized. In this paper, the training data has *no ground-truth labels (positive labels)*, so that the explicit ranking loss can not be adopted for optimization. The proposed KDindex incorporates extra supervision ranking-based signals through a teacher index to capture interactions between queries and items.
> There are also numerous works that use knowledge distillation to improve the performance of hashing-based codes, such as [1], where the ranking information is distilled from a graph-based network to enhance the performance of hashing codes. However, these works rely on the ground-truth labels (user-item interactions) to learn the ranking orders. This is different from our work, where label information is not accessible for learning.
>
> Furthermore, our proposed framework can be adapted to a variety of compressed indexes. The learning to hash method would be instantiated as the student index in the subsequent works to show the generality of the model.
>
> [1] Binarized collaborative filtering with distilling graph convolutional networks. IJCAI 2019.
> [2] BLISS: A Billion scale Index using Iterative Re-partitioning. SIGKDD 2022
>
> 4. [The initialization of the codewords]
>
> > **Comments:** Is it possible to take the help of teacher index to initialize in a more informed manner?
>
> Thank you for your valuable suggestion. A straightforward way is to exploit the neighbor information in the teacher index to aggregate the similar items with the same index. But for the centroids, the size and scale are remarkably different from that in HNSW. This would remain a further problem.
>
> KDindex is initialized with AQ/PQ/OPQ instead of random initialization. KDindex(AQ) represents that the quantization method is warmed up by AQ. The same goes for PQ and OPQ.  We also report that 'To accelerate the training, codebooks are warmed by original quantization methods such as PQ, OPQ, and AQ.' in Appendix D.
>
> 5. [More descriptions abouth Figure 2]
>
> The unit is second which corresponds with the results in [ann-benchmarks](https://ann-benchmarks.com/). KDindex here stands for AQ.
>
> > **Comments:** How can this happen as HNSW is the teacher and KDIndex is the student?
>
> HNSW's performance can be enhanced by extending search latency. In experiment, we take topk neighbor from HNSW whose recall@10 approaching 0.99 at the expense of search time. Thus, the teacher model provides more powerful signals for training student index.
>
> > **Comments:** best recall is achieved by ScaNN though this comes at increased search latency. However, KDIndex plateaus off quickly and increased search latency doesn't seem to help.
>
> We increase the search latency for comparision, which can be referred in Figure 6 in the supplemented global response PDF. According to the figure, KDindex outperforms ScaNN and BLISS.
>
> 6.  [More choices of K]
>
> We conduct experiments on MS MARCO Doc datasets with $K=2$. The experiments are run for 5 times. The Recall@10 is $17.39\pm0.25$ and MRR@10 is $40.10\pm0.21$. Compared with more neighbors, e.g., $K=5$or $K=10$, extremely smaller number performs worse.

---

> > ### Comment · Reviewer_uFQG · 2023-08-18
> > **Reviews and Rebuttal**
> >
> > I've read the reviews and the rebuttal.
> >
> > I thank the authors for their clarification to some of the questions I had asked.
> >
> > I would appreciate if the authors can address some questions in my review that they seem to have missed responding to.

---

> > > ### Author Response · Authors · 2023-08-19
> > > **Response to Reviewer uFQG (1/2)**
> > >
> > > Thanks for your appreciation and reply! We provide more responses.
> > >
> > > 1. [More descriptions about Figure 2]
> > >
> > > > **Comments:** The recall numbers of KDIndex in Figure 2 for both SIFT1M and GIST1M datasets don't match with the recall numbers reported in Table 1. What explains the discrepancy between Tab. 1 and Fig. 2?
> > >
> > >
> > > The discrepancy between Table 1 and Figure 2 results from the different search methods.  In Tab. 1, the methods are quantization-based, and we index and search by asymmetric distance computation (ADC) following typical settings for quantization-based models [1]. In Fig. 2, to compare with other uncompressed methods, the inverted file system combined with the asymmetric distance computation (IVFADC) is adopted to evaluate the performance under different latency times. Specifically, the utilization of IVF is consistent with the settings in ScaNN. We will add these details to the experiment sections.
> > >
> > > [1] Product quantization for nearest neighbor search. TPAMI 2010
> > >
> > > >  **Comments:** Table 1 should report the teacher search index HNSW results to give an idea of the relative performance.
> > >
> > > Refer to response 6 for aAUh
> > >
> > > 2. [Effect of B and W]
> > >
> > > > **Comments:** In Section 4.4, it would be interesting to know the effect of B and W on storage and search latency.
> > >
> > > As for the storage, it would take $B \times \log W$bits to store the index, which is linear with the number of codebooks ($B$) and logarithmic with the number of codewords ($W$).
> > > As for the search latency, we report the search latency of KDindex(PQ) on the SIFT1M dataset based on the ADC retrieval method as follows.
> > >
> > >
> > > | Time (s) | $B=4$   | $B=8$   | $B=16$  | $B=32$  |
> > > | -------- | ------- | ------- | ------- | ------- |
> > > |$W=64$|23.0896|27.0325|35.2343|52.7827|
> > > |$W=128$|23.0791|27.0185|35.2848|53.0061|
> > > |$W=256$|23.1086|27.0060|35.3733|52.8418|
> > > |$W=512$|23.1076|27.0620|35.5124|52.9954|
> > >
> > > The database vector $d$ is represented by $Q(d)$ where $Q(\cdot)$ denotes the quantization function. The distance $d(q,d)$ is approximated by the distance $\tilde d(q,d) = d(q, Q(d)) = d(q, \sum_{b=1}^B c_{w_b}^b )$ where $B$ is the number of codebooks and $c_{w_b}^b$is the w-th codeword of b-th codebook. For each query, the times of distances computation with $W$ centroids are $\frac{D}{B} \times B \times W$. The number of neighbors approximates $\frac{N}{W} \times B$ under balance constraints in the inference stage. Thus, the total computation complexity takes  $D\times N \times B$, which grows linearly with $B$. Thus larger number of the codebooks $B$ corresponds with more latency while larger number of the codewords has little influence about the search latency, which is consistent with the experimental results.
> > >
> > > > **Comments:** On what criteria was B = 8 W = 256 chosen as the optimal hyper-parameter setting? Was this the optimal setting for all the datasets?
> > >
> > > The selection of hyper-parameters $B = 8$ $W = 256$ was determined since that the Bytes for each items' code are 64 bits $(B \times \log W)$. While larger values $B$ and $W$ tend to enhance performance across different scenarios, the resource constraints imposed by storage limitations led us to adopt a consistent configuration for all datasets.
> > >
> > > 3. [Concerns about Balance Strategy]
> > >
> > > > **Comments:** Of the three strategies employed by KDIndex, Balance seems to give least improvement in retrieval performance going by Table 3 (for instance, PQ Recall @10 8.64 vs 8.62). However, Balance adds significant complexity to the training algorithm. It would be good to report and compare the time taken for training KDIndex and w/o Balance in Table 3 to get a better understanding of the tradeoff between incremental improvement in retrieval performance and training complexity.
> > >
> > > The time complexity of the balance strategy based on sinkhorn-knopp is $O(MBW)$ where $M$, $B$ and $W$ denote batch size (the number of queries within the batch), the number of subspaces, and the number of codewords in each codebook. It is acceptable compared to the time complexity of quantization ($O(MBWD)$)
> > > In the training process, the balance strategy cost about 57.49ms for each batch of 64 samples for B=8 and W=256 while the whole batch computation including the forward, backward, and data IO takes about $1.65\pm0.04$ seconds.
> > >
> > > 4. [Details about Table 4]
> > >
> > > > **Comments:** In Table 4, KDIndex refers to which of KDIndex(AQ), KDIndex(PQ), KDIndex(OPQ)?
> > >
> > > KDindex in Table 4 refers to KDIndex(AQ).
> > >
> > > > **Comments:** In Table 4, why is compression for SIFT1M is much lower than GIST1M and other datasets (7 vs 63)?
> > >
> > > It's crucial to note that the dimension of GIST1M vectors is 960, whereas the dimension of SIFT1M vectors is 128. As both datasets are compressed using the same number of bits, the compression ratio is inherently influenced by the dimensionality of the vectors. Higher-dimensional vectors tend to yield higher compression ratios when compressed using the same number of bits.

---

> > > ### Author Response · Authors · 2023-08-19
> > > **Response to Reviewer uFQG (2/2)**
> > >
> > > 5.  [Similarity functions]
> > >
> > > > **Comments:** Why are the similarity functions for MS MARCO different from that for SIFT1M and GIST1M?
> > >
> > > For more general cases, ANNs rely on the L2 distance, such as for  SIFT1M and GIST1M datasets. Under the document retrieval tasks, the similarity functions often take the inner product, which is known as MIPS (Maximum Inner Product Search). KDindex performs well on both settings, which also indicates the good generalization of KDindex over different similarity functions.
> > >
> > > 6. [Additional Complexity]
> > >
> > > > **Comments:** What is the additional time complexity of KDIndex relative to AQ, PQ and OPQ?
> > >
> > > The training ways of KDindex and basic Qutization methods (AQ, PQ and OPQ) are different, therefore the complexity is different in the training phase. The indexing and inference way of KDindex and Qutization methods are consistent and there is no additional time complexity in the index and inference stages. More details are described in Table 8 in the global PDF.
> > >
> > > | Methods          | KDindex (AQ)           | KDindex (OPQ)           | KDindex (PQ)          |
> > > | ---------------- | ---------------------- | ----------------------- | --------------------- |
> > > | Initialization   | $O(MBWD)$              | $O(MWD^2)$              | $O(MWD)$              |
> > > | Training (Full)  | $O(MBWD + MBW + MBWK)$ | $O(MWD^2 + MBW + MBWK)$ | $O(MWD + MBW + MBWK)$ |
> > > | Training (Final) | $O(MBWD)$              | $O(MWD^2)$              | $O(MWD + MBWK)$       |
> > > | Indexing         | $O(NBWD)$              | $O(NBW((D/B)+D^2))$     | $O(NWD)$              |
> > >
> > >
> > >
> > > 7. [Experiments with Larger Datasets]
> > >
> > > > **Comments:** Why haven't 10M SIFT dataset and 5M SIFT dataset been used in the experiments as done by [38]?
> > >
> > > We conduct more experiment results on Yandex DEEP1B which has a larger scale. Please refer to response 3 for LzAv.
> > >
> > > 8. [Discussion of limitations and potential negative societal impact of their work]
> > >
> > > KDindex mainly focuses on distilling knowledge from the teacher index and taking a trade-off between storage and search efficiency. In the future, we will try more student models such as lattice quantization and learning to hash methods to improve accuracy. And we will take labels into account to improve retrieval performance progressively. More details can be found in Appendix E.

---

### Official Review · Reviewer_LzAv · 2023-07-14

**Soundness:** 3 good
**Presentation:** 3 good
**Contribution:** 3 good
**Rating:** 5
**Confidence:** 2

**Summary:**

The paper proposes a method to compress indexes for ANNS by using knowledge distillation. They propose to use a  graph-based index teacher model and use the top-k nearest results obtained from the teacher indexes to act as the supervision signals to optimize the compressed function. The student model is optimized to have the same ranking orders as the teacher models. Different from previous work, they use a differentiable training process that updates the centroids and indexes simultaneously per mini-batch.

**Strengths:**

Vector search is an important direction. This work tries to compress the embedding index, which is a critical task that can save huge storage and also increase the search performance.

The results seem promising. The authors conduct experiments on 4 benchmarks and show that KDindex achieves a 40x index compression ratio, and 2x CPU speedup compared to the non-compressed method (HNSW).

**Weaknesses:**

There are several important questions that are not answered in this paper.

The paper doesn’t mention the training time for the KDindex. In addition to the storage, the indexing time for the index is also important.

One other question is about how to index new documents. The student model training depends on a trained teacher model (HNSW). What if there is a new document added to the index corpus, will the distilled student generalize well for the new document?

Also, it would be interesting to see larger benchmark, e.g. those have 1B entries, and see if the distillation would work.

**Questions:**

Why not reporting the HNSW performance in table 1?

---

> ### Author Rebuttal · Authors · 2023-08-10
>
> 1.  [Training and Indexing time for the KDindex]
>
>     > **Comments**: The paper doesn’t mention the training time for the KDindex. In addition to the storage, the indexing time for the index is also important.
>
>     Thank you for your suggestion, and we conduct an analysis on the complexity of training and indexing time.
>     The whole training process includes the initialization with different conventional quantization-based methods, followed by the proposed learning-based approach. The various quantizers have an impact on the complexity analysis, and thus we discuss them separately.
>     First of all, we describe the notations for clear understanding. Denoted by $D$ item embedding dimension, $B$ the number of subspaces, $W$ the number of centroids in each codebook, $M$ the batch size (the number of queries in each batch), and $K$ the number of neighbors. $N$ is the number of items in the database. As for ScaNN, $K_v$ denotes the number of centroids in VQ ( vector quantization ) and $K_p$ in PQ (  Product Quantization ).
>     In terms of the learning process, we analyze the time complexity of the forward training process during each batch, which encompasses quantization, balance strategy, and distillation loss calculation. The quantization complexity is intricately tied to the specific quantizers employed. The balance strategy exhibits a complexity of $O(MBW)$ and the distillation takes a complexity of $O(MBWK)$. We summarize the complexity with respect to each batch as Table 8 in the global response PDF.
>
>     Considering the training process, the complexity of the balance strategy, i.e., $O(MBW)$, is considerably lower than that of quantization.  The number of neighbors often refers to small numbers, e.g., $K=10$, which are smaller than the dimension of embeddings ($D>100$).
>
>     [1] Sinkhorn distances: Lightspeed computation of optimal transport. NeurIPS 2013
> 2.  [how to index new documents]
>
>     > **Comments**: how to index new documents. The student model training depends on a trained teacher model (HNSW). What if there is a new document added to the index corpus, will the distilled student generalize well for the new document?
>
>     Given that our document corpus consists of over 1 million data points, the addition of a single new document is unlikely to significantly alter the overall data distribution. Unless the new document is notably distinct from the existing data, its influence on the distribution would be minor.
>     As a result, the relationships between documents, both in the teacher index and the student model, tend to remain relatively stable even with the introduction of a new document. Thus, we can assign the new document with appropriate centroids according to the well-trained quantized index, i.e., $\arg\min_{i\in1,2,\dots, W}  \| x - c_i^b \| ^2$ where $i$ is the code (index) in the $b$-th codebook (subspace) for a new document $x$. The new document may share the same index with its similar document.
>     More concretely, only a set of items are retrieved by the teacher index so that the learning of student index is accessible to the subset of items. The other items are assigned to certain index according to the well learned quantization-based index for later inference. The scale of the subset of items is closely related to the number of the neighbors $K$. We vary $K$in the experiment as shown in Table 2 to demonstrate the effectiveness of the well-learned KDindex with subset of items.
>
> 3.  [Larger Dataset with 1B benchmark]
>
>     > **Comments**:  larger benchmark, e.g. those have 1B entries, and see if the distillation would work.
>
>     We add a 1B dataset Yandex Deep1B[1], which is a image descriptor dataset consisting of the projected and normalized outputs from the last fully-connected layer of the GoogLeNet model. The embeddings is pretrained on the Imagenet classification task. Due to the RAM resource constraints, we randomly sample 20M of the datasets.  The details are as follows:
>
>     | Datasets      | #Database  | #Train    | #Test  | Dim  |
>     | ------------- | ---------- | --------- | ------ | ---- |
>     | Yandex Deep1B | 20,000,000 | 7,000,000 | 10,000 | 96   |
>
>
>     \#Train and \#Test here represent the number of queries.
>
>     We compare the KDIndex with different quantizers in the following table, where KDindex outperforms all the original quantizers.
>
>     | Model     | PQ    | KDindex (PQ) | OPQ   | KDindex (OPQ) | AQ    | KDindex (AQ) |
>     | --------- | ----- | ------------ | ----- | ------------- | ----- | ------------ |
>     | Recall@10 | 9.61  | 12.47        | 16.84 | 18.77         | 17.81 | 18.57        |
>     | NDCG@10   | 33.85 | 37.71        | 45.34 | 49.32         | 53.63 | 55.15        |
>
>     [1] Efficient indexing of billion-scale datasets of deep descriptors. CVPR 2016
>
> 4. [Performance of HNSW]
>
>     > **Comments:** Why not reporting the HNSW performance in table 1?
>
>     For Table 1, we perform ADC over these quantization-based methods. Since HNSW is a graph-based method, the comparison with these quantization-based methods is not fair. We provide more details about training.
>
>     For training, We obtain the approximate top-K neighbors from teacher model and the performance is approaching Recall@10 0.99 by adding the search latency. The time and performance used in experimental are as follows:
>
>     | Datasets | SIFT1M | GIST1M | MS MARCO Doc | MS MARCO Passage |
>     | --- | --- | --- | --- | --- |
>     | Recall@10 | 0.9865 | 0.9859 | 0.9292 | 0.9182 |
>     | NDCG@10 | 0.9999 | 0.9999 | N/A | N/A |
>     | MRR@10 | N/A | N/A | 0.9493 | 0.9327 |
>     | Query time (s) | 0.5862 | 1.3082 | 1.4805 | 4.7689 |
>
>     For more comparison about the recall-time performance, please refer to Figure 6 in the global response PDF.

---

> > ### Comment · Reviewer_LzAv · 2023-08-20
> >
> > Thank you for the rebuttal. It's very informative and helpful. I'd like to update my score from 4 to 5.
> >
> > One minor question:
> > `Given that our document corpus consists of over 1 million data points, the addition of a single new document is unlikely to significantly alter the overall data distribution.`
> > >> I think one key issue is that each application will have its own corpus. Do you think the same distilled student would work for different corpus?

---

> > > ### Author Response · Authors · 2023-08-20
> > > **Response to Reviewer LzAv**
> > >
> > > We sincerely appreciate your time in reviewing our work and considering our rebuttal. Your thoughtful reconsideration of the score is truly encouraging, and we are pleased to learn that you found our rebuttal to be informative and helpful.
> > >
> > > As your minor concern, in fact, when a large number of new documents are introduced to the corpus for indexing, we need to retrain the index structure. The distribution of queries in the dataset often differs from  that of documents. By adding a certain number of documents, the discrepancy between these two distributions becomes even larger. The distillation student models are separately learned for different corpora. However, your suggestion provides us  with a valuable direction for future research. Specifically, your suggestion prompts us to explore the potential of a unified model designed for comprehensive indexing, where the challenge lies in achieving alignment across diverse corpora. Thank you once again for your thoughtful feedback.

---

### Author Rebuttal · Authors · 2023-08-10

Dear reviewers,

Thank you for all the time and effort you spent in writing reviews for KDindex. We polish the description and conduct more experiments folloing reviews. The figures (datasets distribution and time-recall curves) and table (time complexity) are attached in PDF.

Authors.

---

> ### Author Response · Authors · 2023-08-16
> **Seeking Reviewer feedback: Kind Reminder for Author/Reviewer Discussion Phase**
>
> Dear Reviewers,
>
> We would like to express our sincere gratitude for all the help you have provided in reviewing our work. Your feedback has been invaluable in helping us improve our submission. We have carefully prepared a response of each of your reviews and hope it adequately addressed your concerns.
>
> As the deadline for the author/reviewer (Aug 21st) is fast approaching. and we have yet to receive any feedback, we hope to kindly request your help once again. We understand that you may have busy schedules and may still be reviewing our response, but we would greatly appreciate it if we could have further discussions with you to ensure that our submission meets the high standards of Neurips. Your feedback is precious to us, and we are eager to work with you to make any necessary revisions.
>
> Thank you again for your time and consideration.
>
> Best regards.
>
> Authors.

---

### Decision · Program_Chairs · 2023-09-21

**Decision:**

Accept (poster)

**Comment:**

The paper proposes a method to compress indexes for ANNS by using knowledge distillation. All the reviewers seem excited about the proposal and there was a very healthy discussions where most concerns from the reviewers were addressed.  Overall, all the reviewers agree that the proposed contribution is worthy of publication.